# Who pays and how much? A cross-sectional study of out-of-pocket payment for modern contraception in Kenya

Emma Radovich,[1] Mardieh L Dennis,[1] Edwine Barasa,[2] Francesca L Cavallaro,[1] Kerry LM Wong,[1] Josephine Borghi,[3] Caroline A Lynch,[1] Mark Lyons-Amos,[1] Timothy Abuya,[4] Lenka Benova[1,5]

[1]Faculty of Epidemiology & Population Health, London School of Hygiene and Tropical Medicine, London, UK
[2]Health Economics Research Unit, KEMRI-Wellcome Trust Research Programme, Nairobi, Kenya
[3]Faculty of Public Health & Policy, London School of Hygiene and Tropical Medicine, London, UK
[4]Population Council Kenya, Nairobi, Kenya
[5]Department of Public Health, Instituut voor Tropische Geneeskunde, Antwerpen, Belgium

**Correspondence to**
Emma Radovich;
emma.radovich@lshtm.ac.uk

## ABSTRACT

**Objectives** Out-of-pocket (OOP) payment for modern contraception is an understudied component of healthcare financing in countries like Kenya, where wealth gradients in met need have prompted efforts to expand access to free contraception. This study aims to examine whether, among public sector providers, the poor are more likely to receive free contraception and to compare how OOP payment for injectables and implants—two popular methods—differs by public/private provider type and user's sociodemographic characteristics.

**Design, setting and participants** Secondary analyses of nationally representative, cross-sectional household data from the 2014 Kenya Demographic and Health Survey. Respondents were women of reproductive age (15–49 years). The sample comprised 5717 current modern contraception users, including 2691 injectable and 1073 implant users with non-missing expenditure values.

**Main outcome** Respondent's self-reported source and payment to obtain their current modern contraceptive method.

**Methods** We used multivariable logistic regression to examine predictors of free public sector contraception and compared average expenditure for injectable and implant. Quintile ratios examined progressivity of non-zero expenditure by wealth.

**Results** Half of public sector users reported free contraception; this varied considerably by method and region. Users of implants, condoms, pills and intrauterine devices were all more likely to report receiving their method for free (p<0.001) compared with injectable users. The poorest were as likely to pay for contraception as the wealthiest users at public providers (OR: 1.10, 95% CI: 0.64 to 1.91). Across all providers, among users with non-zero expenditure, injectable and implant users reported a mean OOP payment of Kenyan shillings (KES) 80 (US$0.91), 95% CI: KES 78 to 82 and KES 378 (US$4.31), 95% CI: KES 327 to 429, respectively. In the public sector, expenditure was pro-poor for injectable users yet weakly pro-rich for implant users.

**Conclusions** More attention is needed to targeting subsidies to the poorest and ensuring government facilities are equipped to cope with lost user fee revenue.

### Strengths and limitations of this study

► A major strength of the study is that it is the first to our knowledge to use nationally representative data from a low-income country to examine out-of-pocket payment for modern contraception.
► Another strength is the transparency in the classification of family planning providers, handling of outliers and appropriate adjustments for complex survey design.
► One limitation of the study is the reliance on self-reported cost data from current users of modern contraception and the inability to compare this with costs to women who discontinued or eschewed use of modern contraception.

## BACKGROUND

Achieving Universal Health Coverage—including for family planning (FP) services—demands attention to financial protection. Consideration of user fees is particularly important in countries like Kenya, where out-of-pocket (OOP) payments form a substantial proportion of healthcare financing.[1 2] In Kenya, unmet need for FP is highest among the poor, with a documented 8–14 percentage point increase in modern contraception use with each increase in household wealth quintile.[3] A study in Kenya and India found that poor households spend a significantly higher proportion of their income on reproductive healthcare (including FP), with the poorest households in Kenya spending 10 times the proportion spent by the least poor.[4] Many government financial protection policies focus on inpatient events where healthcare expenditure is likely to be catastrophic, yet the greater frequency of outpatient expenses—including for contraceptive services, which affect women in particular—can also push

households into poverty[5] or reduce care-seeking among the poor.[3 6]

A systematic review on the relationship between user fees and FP use in low-income and middle-income countries (LMICs) was inconclusive, although some included studies suggested that young people and the poor were more sensitive to price increases than wealthier or less marginalised groups.[7] Cost is rarely cited as the reason for non-use of modern contraception among women in need (those wishing to delay or avoid pregnancy) in Demographic and Health Surveys (DHS).[8 9] Yet, focus groups in Nyanza Province, Kenya found that the poor identify high cost of services as a barrier to FP care, both in opportunity costs associated with seeking care and direct fees for services,[3] suggesting that for some individuals, cost can impact FP access.

Kenya has used various financing mechanisms to support increased access to FP and reproductive health services.[3 10] A 2004 policy, commonly known as the '10/20 policy', abolished user fees in primary care facilities; instead government dispensaries and health centres were allowed to charge a registration fee of 10 or 20 Kenyan shillings (KES) (approximately US$0.11 and US$0.23), with the poor exempted from payment.[11 12] Public hospitals could continue charging fees to users under a cost sharing policy. Yet fee waiver implementation and identification of eligible individuals was left to the discretion of actors at the community and facility level. Despite the 10/20 policy, many FP clients in government facilities reported paying additional 'hidden fees' for the consultation, medical tests or equipment and the contraceptive commodity.[3] A 2010 health facility survey found that approximately 70% of government facilities providing FP charged user fees for services.[13] A 2009 study found low community knowledge of the 10/20 policy and qualifying exemptions.[11] However, as of June 2013, all fees at public outpatient primary care facilities (dispensaries and health centres) were eliminated,[12] and FP services are intended to be provided for free at public facilities.[14 15] The extent to which users currently receive free FP services from public outpatient primary care facilities is unknown, and similar to the 10/20 policy, implementation of the June 2013 policy for free FP services may vary, for example, by facility type, geographic region or client characteristics.

Efforts to achieve universal coverage for reproductive health have led to increasing calls by donors and others for a 'total market approach' in considering the different contributions of public and private providers. In this approach, government-subsidised or otherwise subsidised services are targeted to meet the needs of the poor while individuals with the ability to pay are indirectly encouraged to seek FP services from commercial or unsubsidised private providers.[3 5 16 17] Kenya's changing fee policies within the public sector and the country's growing private sector, which now owns half of all health facilities,[12] raise questions about where individuals, especially the poor, seek FP and what this means for their OOP payment for modern contraception. Little is known about OOP payment to obtain modern contraception in sub-Saharan Africa, and in Kenya in particular, and how this varies by provider type. In the context of limited resources to expand FP access,[18] it is important to understand the burden of user fees—who pays and how much—and the degree to which vulnerable groups are served by current efforts to provide affordable care.

This paper aims to address these knowledge gaps by describing FP sources by user's wealth in Kenya, examining, among public sector users, who receives free FP services, and comparing how payment for injectables and implants—the two most commonly used methods—differs by FP provider type and the user's sociodemographic characteristics.

## METHODS

### Data source

We used data from the most recent Kenya DHS (2014), a nationally representative, cross-sectional household survey of women age 15–49 years with a multilevel cluster sampling design. A detailed description of the survey sampling can be found in the DHS report.[19] Interviews were administered between May and October 2014. Our analysis includes women in half of the randomly selected households who were administered the long-version Woman's Questionnaire (unweighted n=14 741), which included a question on the amount paid for the respondent's current contraceptive method.[19] Respondents were not asked for the reason for the payment.

### Study populations

We examined data from three populations of women: (1) current users of modern contraception, based on the Hubacher and Trussell definition of modern methods[20]; (2) users of intrauterine device (IUD), implant, injectable, pill and male condom as these users were asked to self-report the total amount paid to obtain their method (the combined cost of the commodity and any consultation fees) during their most recent (re)supply visit and (3) users of injectable and implant, where estimates of OOP payment refer to a single quantity of the contraceptive, as users can receive only one 'dose' during insertion or resupply. Respondents with missing or 'do not know' expenditure values accounted for 4.4% of all users in group two, and <1% of injectable and implant users, and were excluded from analysis.

### Definitions

We classified women's self-reported most recent source of modern FP into seven provider categories: 1) government hospital; 2) government health centre; 3) government dispensary; 4) private facility, a constructed category comprising DHS response options of private hospital/clinic and private nursing/maternity home; 5) non-governmental organisation (NGO)/faith-based facility; 6) pharmacy/chemist and 7) other, a constructed category of the response options: shop,

mobile clinic, friend/relative, other, community health worker, community-based distributor and other private medical. We defined the public sector to be government-provided services (categories 1–3) and non-public providers to be categories 4–7. We consider public primary care providers to be categories 2 and 3. Less than 1% of all current modern contraceptive users were missing the source of their method and were excluded from analysis.

We examined three measures of the respondents' socioeconomic status: household wealth quintiles derived by the DHS from household assets,[21] urban/rural residence and three levels of educational attainment: less than primary school (respondents with no education and those who started but did not complete primary school), less than secondary school (respondents with complete primary or incomplete secondary school) and secondary+ (respondents with complete secondary or some higher education). We used DHS categories for respondent's current marital status (never, currently or formerly in union) and grouped respondents by their current age: <20, 20–29 and 30+ years. Kenya is administratively divided into 47 counties; however, the variable for OOP payment for contraception in the 2014 Kenya DHS was intended to provide representative estimates for the national level, for urban and rural areas and for the eight regions (former provinces).[19]

### Analysis of free or 'registration fee only' FP in the public sector

We limited analysis of free FP to users whose most recent source of the method was a public sector provider. We include both categories of public primary care providers (subject to the June 2013 abolishment of fees) and government hospitals as a point of comparison. Adjusted Wald tests were performed to compare proportions reporting free FP by facility type and user characteristics. Bivariable and multivariable logistic regression was used to examine predictors, such as wealth quintile, facility type and region, of receiving free FP from public primary care facilities, as indicated under the 2013 policy.

Users of long-acting methods like IUD and implant could report OOP payment based on FP consultations before the June 2013 abolishment of fees. Thus, estimates of free FP for IUD and implant were further disaggregated by whether the method was obtained before or after June 2013 based on the respondent's self-reported month and year of initiating use of the method.

Facility-level implementation of the June 2013 policy abolishing all fees at public primary care facilities may not have been immediate. As such, we additionally examined the proportion of users who reported paying up to 10 KES or 20 KES at a government dispensary or health centre, respectively, referring to these users as paying 'registration fees only' consistent with the former 10/20 policy, although respondents did not indicate the reason for the charge.

### Analysis of OOP payment for injectable and implant

Prior to analysis, we assessed the data for improbable values and recoded observations to missing if reported expenditure was >10 times the 95th percentile (six observations). Among injectable and implant users reporting non-zero cost, we described the patterns of OOP expenditure, reporting mean and median values. We conducted sensitivity analyses to ensure the robustness of our results, comparing results from multiple methods for dealing with outliers[22]; results did not differ substantially (online supplementary table 1). For this analysis, observations >2 SD from the mean (2.7% and 2.1% of injectable and implant users, respectively) were recoded to be equal to the mean. Simple linear regression and marginal effects were used to compare means between providers and user characteristics. We additionally present estimates of OOP payment converted from KES to US dollars based on 1 KES to US$0.0114 conversion rate for the midpoint of fieldwork in July 2014[23] (online supplementary tables 1–2).

### Equity of OOP payment for injectable and implant

Quintile ratios were used to measure the progressiveness of OOP payments for injectables and implants overall and within the public sector. This measure of equity in expenditure assumes that individuals in the lowest wealth quintile have less capacity to pay and thus if they spend the same or more as those in the highest quintile, this represents a greater proportion of income and constitutes regressive spending.[24] Quintile ratios were calculated by comparing mean expenditure in the wealthiest and poorest wealth quintiles and testing for differences using an adjusted Wald-type test of non-linear hypotheses based on the delta method, attributing significance at a 95% confidence level.[4 24 25] We define expenditure as weakly pro-rich if there was no significant difference in mean payment between the poorest and wealthiest users and strongly pro-rich if the poorest users paid significantly more than the wealthiest users (quintile ratio <1).[4 24]

All analysis used women's individual sampling weights and SE adjustment to account for complex survey design. Analyses were conducted in Stata/SE V.14 (StataCorp, College Station, Texas, USA).

### Patient and public involvement statement

Patients and the public were not involved in this secondary data analysis.

## RESULTS

A total of 5717 (weighted n) modern contraceptive users with non-missing provider data were included in our analysis sample.

### Methods and sources of family planning

Among all current modern FP users, the wealthiest quintile had the broadest mix of methods, with no single method accounting for more than a third of modern FP users (figure 1A). In contrast, method mix among users

 

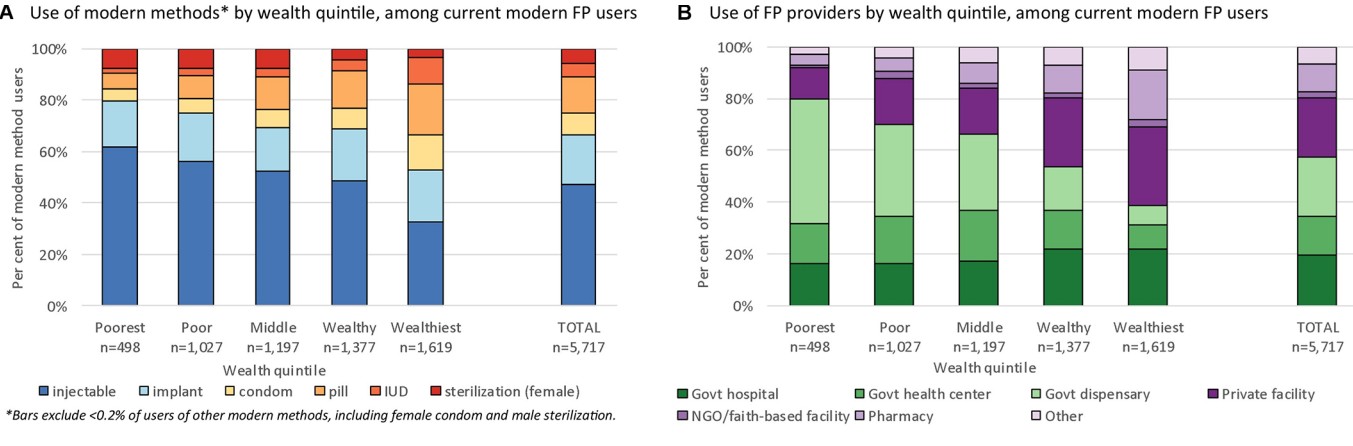

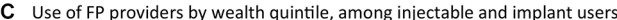

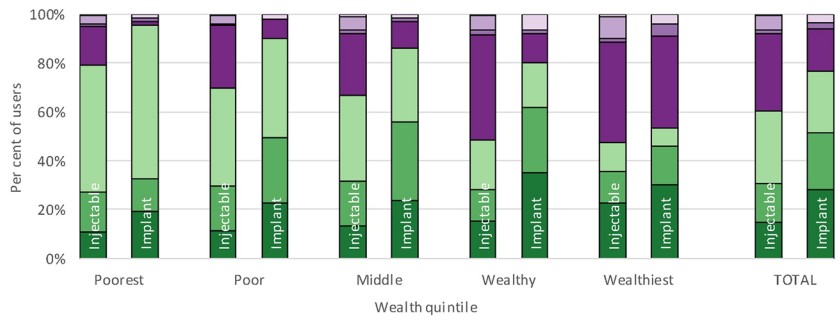

**Figure 1**  Method mix and provider use by wealth quintile among current modern family planning (FP) users.

in the three poorer quintiles was dominated by inject-ables, which accounted for more than half of methods used. While injectables and implants were the two most popular methods for all users, this was particularly true for the poorest users, where these two methods accounted for nearly 80% of all modern methods used.

The wealthiest contraceptive users also reported a broader mix of providers (figure 1B). Among the poorest users, 80.0% reported a public sector source. Public provider use decreased steadily and use of private facil-ities and pharmacies increased with increasing wealth quintile. The wealthiest users reported the largest use of private facilities (30.5%) and pharmacies (18.7%). Among injectable users, public sector providers were the most-used source for the three poorest quintiles, with a clear decline in government dispensary use with increasing wealth (figure 1C). The vast majority of implants in the four poorer wealth quintiles were sourced from public providers, and there was a dramatic increase in use of private facilities for implants in the fifth, wealthiest quin-tile (online supplementary table 3 shows the distribution of all modern methods by provider type).

### Free family planning

Users of injectable, implant, pill, condom and IUD were asked to self-report the total amount paid to obtain their method. Overall, 51.1% of public sector users reported obtaining their modern FP method for free at their most

recent visit (table 1). This varied by method used: >90% of condom users compared with 40.7% of injectable users reported free FP. Across the three levels of facilities, 50.1% of government hospital, 56.2% of government health centre and 48.5% of government dispensary users reported free FP, with some evidence of a difference by facility type (p=0.048). The percentage of women obtaining free FP in public facil-ities differed only slightly by respondent's wealth quintile, urban/rural residence, education level or age group. The proportion of users reporting free FP varied considerably by region, with 39.4% of Rift Valley residents compared with 76.6% of Nairobi residents reporting free contracep-tion. Additionally, 1.3% (95% CI: 0.9% to 2.1%) of users of government health centres and dispensaries reported paying a 'registration fee only' amount under the former 10/20 policy (results not shown). There was no difference by user's wealth quintile.

Among non-public sector providers (results not shown), 10.9% of private facility users and <1% of pharmacy users reported free FP. Of the limited number of users of NGO/ faith-based facilities (n=91), 30.9% reported obtaining their contraceptive method for free.

Online supplementary table 4 shows the proportion of IUD and implant users receiving free FP from public sector providers among users initiating the method before and after the June 2013 fee abolishment. Among implant users, the proportion receiving free FP from government

**Table 1** Among public sector providers, proportion reporting free family planning by modern method users' sociodemographic characteristics

| | Government hospital (n=929) | Government health centre (n=815) | Government dispensary (n=1267) | Total public (n=3011) |
|---|---|---|---|---|
| Overall (95% CI) | 50.1 (45.9 to 54.3) | 56.2 (50.9 to 61.4) | 48.5 (45.0 to 52.1) | 51.1 (48.5 to 53.7) |
| **Method** | | | | |
| Injectable | 38.4 (32.4 to 44.8) | 46.0 (39.3 to 52.9) | 39.2 (34.8 to 43.7) | 40.7 (37.5 to 44.1) |
| Implant | 55.0 (47.1 to 62.6) | 63.0 (53.9 to 71.1) | 61.5 (54.6 to 67.9) | 59.6 (54.8 to 64.2) |
| Pill | 68.7 (53.8 to 80.6) | 66.0 (50.6 to 78.6) | 61.1 (50.0 to 71.3) | 64.7 (57.1 to 71.6) |
| Condom | 90.8 (71.0 to 97.5) | 92.8 (77.4 to 98.0) | 97.4 (83.7 to 99.6) | 93.6 (85.0 to 97.3) |
| Intrauterine device | 49.6 (38.6 to 60.7) | 75.4 (57.8 to 87.3) | 73.4 (57.1 to 85.1) | 60.7 (52.8 to 68.1) |
| **Wealth quintile** | | | | |
| Poorest | 61.7 (48.7 to 73.1) | 54.1 (41.3 to 66.5) | 46.1 (39.2 to 53.1) | 50.2 (44.4 to 55.9) |
| Poor | 51.2 (41.0 to 61.4) | 51.7 (42.2 to 61.1) | 44.3 (38.4 to 50.4) | 47.6 (43.0 to 52.3) |
| Middle | 43.2 (35.5 to 51.2) | 52.4 (42.6 to 62.0) | 48.9 (41.3 to 56.5) | 48.6 (43.6 to 53.7) |
| Wealthy | 51.8 (43.6 to 60.0) | 57.1 (47.3 to 66.3) | 54.4 (45.9 to 62.7) | 54.2 (48.6 to 59.7) |
| Wealthiest | 49.6 (42.0 to 57.2) | 66.9 (55.0 to 76.9) | 53.9 (39.8 to 67.4) | 54.9 (48.7 to 61.0) |
| **Residence** | | | | |
| Urban | 49.1 (43.6 to 54.7) | 66.0 (56.8 to 74.2) | 56.1 (47.6 to 64.3) | 55.2 (50.6 to 59.7) |
| Rural | 51.6 (45.1 to 58.0) | 50.9 (44.7 to 57.0) | 46.8 (42.9 to 50.7) | 48.8 (45.8 to 51.9) |
| **Educational attainment** | | | | |
| Less than primary | 56.7 (48.3 to 64.7) | 55.1 (47.0 to 62.9) | 47.6 (42.5 to 52.6) | 51.7 (47.8 to 55.6) |
| Less than secondary | 46.0 (40.0 to 52.0) | 56.6 (49.3 to 63.6) | 48.9 (43.8 to 53.9) | 49.9 (46.4 to 53.5) |
| Secondary+ | 51.3 (43.0 to 59.4) | 57.1 (46.9 to 66.6) | 49.8 (40.9 to 58.7) | 52.5 (47.0 to 58.0) |
| **Age group (years)** | | | | |
| <20 | 41.3 (19.4 to 67.4) | 61.3 (40.2 to 78.9) | 60.4 (42.8 to 75.7) | 55.4 (43.4 to 66.9) |
| 20–29 | 45.8 (40.0 to 51.7) | 55.3 (48.1 to 62.4) | 42.9 (37.9 to 48.1) | 47.0 (43.5 to 50.6) |
| 30+ | 54.2 (48.0 to 60.3) | 56.5 (49.8 to 63.0) | 52.7 (47.8 to 57.5) | 54.2 (50.7 to 57.7) |
| **Region*** | | | | |
| Central | 53.1 (43.5 to 62.4) | 64.0 (50.7 to 75.4) | 60.2 (47.8 to 71.4) | 58.6 (51.3 to 65.5) |
| Coast | 70.3 (58.9 to 79.7) | 81.2 (69.6 to 89.0) | 62.4 (52.4 to 71.4) | 69.1 (62.1 to 75.2) |
| Eastern | 35.2 (26.3 to 45.2) | 40.4 (29.5 to 52.3) | 44.5 (36.9 to 52.4) | 41.5 (36.0 to 47.2) |
| Nairobi | 70.4 (54.1 to 82.7) | 76.0 (55.3 to 89.0) | –† | 76.6 (63.4 to 86.0) |
| Nyanza | 59.4 (49.8 to 68.3) | 55.6 (44.2 to 66.4) | 37.0 (30.0 to 44.7) | 49.0 (43.1 to 55.0) |
| Rift Valley | 30.9 (24.2 to 38.5) | 48.0 (37.3 to 58.9) | 42.8 (36.7 to 49.1) | 39.4 (35.1 to 43.8) |
| Western | 60.1 (43.6 to 74.5) | 46.6 (34.2 to 59.4) | 50.0 (39.8 to 60.3) | 50.6 (43.5 to 57.7) |

*Due to the very low modern contraceptive prevalence (<5%), results for the North Eastern region are not presented.
†No respondents reported this provider.

health centres was similar between the two initiation periods and increased in the later period among users of government hospitals and dispensaries, although CIs overlap. Among IUD users, the proportion receiving free care was slightly lower across all three public provider categories in the later initiation period, but differences were not statistically significant.

Table 2 shows the results of bivariable and multivariable analysis of receiving free modern FP among users of public primary care facilities, which were subject to the June 2013 fee abolishment policy. There were no differences by wealth quintile in the odds of obtaining free contraception after adjusting for method, provider type and user characteristics. Users of implants, condoms, pills and IUDs were all more likely to report receiving their method for free (p<0.001) compared with injectable users, and this relationship remained after adjusting for provider and user characteristics. Users in all regions had lower odds of free contraception compared with Nairobi, except Coast region (where it was not significantly different).

**Table 2** Unadjusted and adjusted ORs from logistic regression analysis of reporting free family planning services from government primary care providers among modern method users

| Variables | Modern method users using public primary care providers (n=2079) | |
| --- | --- | --- |
| | Unadjusted OR (95% CI) | Adjusted OR (95% CI) |
| Wealth quintile | | |
| Poorest | 0.58* (0.37 to 0.91) | 1.10 (0.64 to 1.91) |
| Poor | 0.55** (0.36 to 0.85) | 1.20 (0.71 to 2.03) |
| Middle | 0.64* (0.41 to 0.99) | 1.25 (0.74 to 2.11) |
| Wealthy | 0.79 (0.51 to 1.24) | 1.16 (0.67 to 2.01) |
| Wealthiest | Ref | Ref |
| Provider | | |
| Government health centre | Ref | Ref |
| Government dispensary | 0.74* (0.57 to 0.95) | 0.95 (0.74 to 1.22) |
| Method | | |
| Injectable | Ref | Ref |
| Implant | 2.32*** (1.78 to 3.02) | 2.15*** (1.62 to 2.86) |
| Condom | 29.87*** (9.84 to 90.66) | 35.29*** (11.42 to 109.05) |
| Pill | 2.39*** (1.63 to 3.52) | 2.27*** 1.56 to 3.28) |
| Intrauterine device | 4.14*** (2.26 to 7.56) | 3.90*** (2.06 to 7.36) |
| Residence | | |
| Urban | Ref | Ref |
| Rural | 0.58*** (0.43 to 0.79) | 0.83 (0.60 to 1.14) |
| Region | | |
| Central | 0.36* (0.14 to 0.93) | 0.31* (0.10 to 0.93) |
| Coast | 0.48 (0.18 to 1.24) | 0.53 (0.19 to 1.53) |
| Eastern | 0.17*** (0.07 to 0.42) | 0.18** (0.06 to 0.52) |
| Nairobi | Ref | Ref |
| Nyanza | 0.18*** (0.07 to 0.44) | 0.17** (0.06 to 0.50) |
| Rift Valley | 0.17*** (0.07 to 0.43) | 0.19** (0.07 to 0.53) |
| Western | 0.21** (0.08 to 0.53) | 0.21** (0.07 to 0.62) |
| Age group (years) | | |
| <20 | 1.31 (0.75 to 2.28) | 1.58 (0.85 to 2.92) |
| 20–29 | 0.76* (0.61 to 0.95) | 0.85 (0.67 to 1.08) |
| 30+ | Ref | Ref |
| Marital status | | |
| Never in union | 0.93 (0.62 to 1.40) | 0.80 (0.47 to 1.36) |
| Currently in union | Ref | Ref |
| Formerly in union | 1.38† (0.96 to 1.98) | 1.27 (0.88 to 1.83) |

*p<0.05, **p<0.01, ***p<0.001 (differences between the category and the reference category are significant).
†p<0.1 (differences between the category and the reference category are marginally significant).

## Out-of-pocket payment for injectables and implants

Among injectable and implant users reporting greater than zero OOP payment to obtain the method from their most recent provider (both sectors combined), the mean cost was KES 80 (US$0.91) (95% CI: KES 78 to 82) for injectable and KES 378 (US$4.31) (95% CI: KES 327 to 429) for implant (table 3); 1.7% of injectable and 1.5% of implant users reported paying amounts consistent with registration fees only (<KES 20). OOP payment varied, particularly for implant, by source of the method, with some private facility users reporting very high costs. Injectable users of public sector providers reported a median cost of KES 50, whereas the median cost was twice that (KES 100) for those accessing private facilities or pharmacies. Among implant users, those accessing public sector sources reported a median cost of KES 200, compared with a median cost of KES 503 among those using private facilities.

When assessed by user characteristics, mean OOP payment for both injectables and implants varied significantly by user's wealth, residence, education level and region, but not by user's age (table 4). Urban and Nairobi residents paid more for both methods; this was particularly notable for implant users in Nairobi, where mean cost was more than twice that of implant users in Western or Nyanza regions. Mean and median cost did not increase linearly with increasing wealth quintile. For injectable users, median cost in the poorest three quintiles was KES 70 compared with KES 100 in the two wealthiest quintiles. For implant users, median cost of KES 500 in the wealthiest quintile was more than twice the median cost of KES 200 in the four poorer quintiles. The overall quintile ratio for all providers comparing mean cost in the wealthiest quintile with the poorest quintile was 1.3 (p<0.001) for injectable and 1.8 (p=0.007) for implant, indicating strong evidence of pro-poor OOP payment for both methods. Among public sector users, the quintile ratio was 1.2 (p=0.033) for injectable indicating pro-poor expenditure, and 0.90 (p=0.660) for implant (table 5), indicating weakly pro-rich expenditure (no difference in mean cost between the quintiles) for public sector implant users.

## DISCUSSION

This is the first study to our knowledge to use nationally representative household data from an LMIC to examine equity of OOP payment for FP, comparing differences in cost with users accessing public and non-public providers. The wealthiest FP users in Kenya used a greater mix of modern methods and providers compared with the poorest users, and use of non-public providers increased with increasing wealth. Despite Kenya's national policy to offer free FP services at public primary care facilities, we found only half of modern method users reported obtaining their method at no cost from government providers, with little variation by facility type. There were no differences by user's socioeconomic position. Among injectable and implant users reporting OOP expenditure, there were considerable differences by source of the method. Consistent with a previous study of FP users in urban Kenya,[26] we found private facility and pharmacy

**Table 3** Summary of out-of-pocket payment (in KES) for injectable and implant users among users with non-zero expenditure, by most recent provider of the contraceptive method

| | Government hospital | Government health centre | Government dispensary | Total public | Private facility | NGO/ faith-based facility | Pharmacy/ chemist | Other* | Total |
|---|---|---|---|---|---|---|---|---|---|
| **Injectable** | | | | | | | | | |
| n | 247 | 225 | 490 | 962 | 821 | 28 | 148 | 17 | 1976 |
| Mean cost | KES 72 | KES 66 | KES 63 | KES 66 | KES 94 | KES 75 | KES 95 | KES 93 | KES 80 |
| SD | 33.38 | 28.37 | 28.78 | 30.10 | 24.33 | 26.37 | 24.42 | 24.56 | 30.63 |
| 25th percentile | KES 50 | KES 50 | KES 50 | KES 50 | KES 80 | KES 50 | KES 80 | KES 100 | KES 50 |
| 50th percentile (median) | KES 50 | KES 50 | KES 50 | KES 50 | KES 100 | KES 70 | KES 100 | KES 100 | KES 100 |
| 75th percentile | KES 100 | KES 100 | KES 100 | KES 100 | KES 100 | KES 87 | KES 100 | KES 100 | KES 100 |
| Reporting registration fees only | 7.5% | 5.3% | 0.8% | 3.6% | | | | | 1.7% |
| **Implant** | | | | | | | | | |
| n | 136 | 94 | 102 | 332 | 130 | 11 | – | 3 | 477 |
| Mean cost | KES 305 | KES 255 | KES 208 | KES 261 | KES 655 | KES 564 | | KES 544 | KES 378 |
| SD | 295.01 | 221.51 | 142.92 | 238.98 | 441.62 | 388.58 | | 534.17 | 359.25 |
| 25th percentile | KES 200 | KES 100 | KES 100 | KES 100 | KES 300 | KES 200 | | KES 100 | KES 200 |
| 50th percentile (median) | KES 200 | KES 200 | KES 200 | KES 200 | KES 503 | KES 800 | | KES 100 | KES 200 |
| 75th percentile | KES 300 | KES 300 | KES 200 | KES 300 | KES 1000 | KES 800 | | KES 1000 | KES 500 |
| Reporting registration fees only | 1.7% | 5.0% | 0.0% | 2.1% | | | | | 1.5% |

*Includes Demographic and Health Survey response options: mobile clinic and other private medical.
1 KES=US$0.0114.
KES, Kenyan shillings; NGO, non-governmental organisation.

users, unsurprisingly, reported higher expenditures than users of public facilities. Unfortunately, due to very small sample sizes (<30 users), OOP payment by users of NGO/faith-based facilities remains unclear, although there is some indication that costs may be higher than among public sector providers. Greater use of higher cost, non-public providers by the wealthiest users contributed to overall pro-poor expenditure, with both injectable and implant users in the wealthiest quintile paying significantly more than their counterparts in the poorest quintile.

A 'total market approach' to FP includes efforts to target government subsidies to the poorest contraceptive users and indirectly nudge wealthier users to seek FP from non-public providers. Evidence from this study suggests that market forces appear to be working to encourage greater use of non-public providers by the wealthiest users, although nearly 40% of FP users in the wealthiest quintile still sourced their method from the public sector. However, while the poorest users obtained their methods overwhelmingly from public providers they were equally likely to pay for FP as users in the wealthiest quintile, suggesting the potential for better targeting of free services to ensure the national pro-poor strategy of removing user fees for FP in public primary care facilities is reaching recipients most in need.

The Kenyan government faces the challenge of both meeting targets to reduce unmet need for FP[14] and ensuring all women, including the poor, have choice in FP methods and providers. Recent attempts to expand access to long-term methods, like implants, in Kenya have focused on expanding the range of providers available to the poor through vouchers. In 2005, Kenya launched a pilot system in five districts that enabled individuals below the poverty threshold to purchase vouchers for long-term or permanent contraceptive methods, which could be redeemed at a variety of public, private for-profit and private not-for-profit providers.[27] The FP voucher programme received criticism concerning the limited uptake of the scheme[2 28] and lack of demand generation activities. Some also suggested that the FP voucher fee of KES 100 (approximately US$1.25) was still relatively costly for the poorest users,[3] although this is half the reported median cost (KES 200) for implant in the public sector in our study. Studies in Kenya have found many women express a preference for or high satisfaction with FP services at private sector facilities.[29–31] Initiatives to expand the range of affordable providers offering high-quality care and a range of contraceptives are still important components of ensuring FP access and choice.

Respondent's region was significantly associated with differences in reporting free FP and the amount paid

**Table 4**  Out-of-pocket (OOP) payment (in KES) for injectable and implant across all provider types among users with non-zero expenditure by sociodemographic characteristics

| | Injectable | | | | Implant | | | |
|---|---|---|---|---|---|---|---|---|
| | n | Median | Mean (95% CI) | Quintile ratio† | n | Median | Mean (95% CI) | Quintile ratio† |
| **Wealth quintile** | | | | | | | | |
| Poorest | 209 | KES 70 | KES 71 (66 to 77) | | 29 | KES 200 | KES 294 (165 to 422) | |
| Poor | 417 | KES 70 | KES 71 (67 to 74) | | 89 | KES 200 | KES 244 (212 to 274) | |
| Middle | 459 | KES 70 | KES 76 (73 to 79) | p<0.001 | 81 | KES 200 | KES 266 (223 to 309) | p<0.001 |
| Wealthy | 516 | KES 100 | KES 83 (80 to 87) | | 101 | KES 200 | KES 357 (248 to 465) | |
| Wealthiest | 379 | KES 100 | KES 96 (91 to 101) | 1.3 (p<0.001) | 177 | KES 500 | KES 522 (415 to 629) | 1.8 (p=0.007) |
| **Residence** | | | | | | | | |
| Urban | 790 | KES 100 | KES 91 (88 to 94) | p<0.001 | 230 | KES 200 | KES 455 (364 to 545) | p=0.005 |
| Rural | 1191 | KES 70 | KES 73 (71 to 75) | | 246 | KES 300 | KES 306 (258 to 355) | |
| **Educational attainment** | | | | | | | | |
| Less than primary | 615 | KES 80 | KES 75 (72 to 78) | | 115 | KES 200 | KES 340 (255 to 425) | |
| Less than secondary | 915 | KES 100 | KES 80 (77 to 84) | p<0.001 | 202 | KES 200 | KES 295 (244 to 346) | p=0.004 |
| Secondary+ | 451 | KES 100 | KES 87 (83 to 91) | | 160 | KES 300 | KES 510 (394 to 626) | |
| **Age group (years)** | | | | | | | | |
| <20 | 77 | KES 100 | KES 81 (74 to 87) | | 9 | KES 400 | KES 307 (171 to 442) | |
| 20–29 | 1030 | KES 87 | KES 80 (77 to 83) | p=0.928 | 226 | KES 200 | KES 369 (304 to 433) | p=0.594 |
| 30+ | 874 | KES 100 | KES 80 (77 to 82) | | 242 | KES 200 | KES 389 (307 to 472) | |
| **Region*** | | | | | | | | |
| Central | 207 | KES 100 | KES 90 (86 to 95) | | 87 | KES 300 | KES 396 (304 to 488) | |
| Coast | 125 | KES 100 | KES 82 (73 to 92) | | 11 | KES 200 | KES 379 (119 to 639) | |
| Eastern | 425 | KES 80 | KES 77 (73 to 81) | | 67 | KES 300 | KES 414 (333 to 495) | |
| Nairobi | 183 | KES 100 | KES 101 (91 to 111) | p<0.001 | 51 | KES 503 | KES 704 (374 to 1034) | p<0.001 |
| Nyanza | 315 | KES 50 | KES 72 (68 to 77) | | 66 | KES 200 | KES 255 (185 to 324) | |
| Rift Valley | 492 | KES 87 | KES 80 (76 to 84) | | 129 | KES 200 | KES 358 (275 to 440) | |
| Western | 232 | KES 70 | KES 69 (64 to 74) | | 64 | KES 200 | KES 226 (183 to 270) | |

*Due to the very low modern contraceptive prevalence (<5%), results for the North Eastern region are not presented.
†Ratio of mean OOP expenditure comparing the wealthiest users with the poorest users. Adjusted Wald-type test based on the delta method was used to test for significance.
1 KES=US$0.0114.
KES, Kenyan shillings.

**Table 5** Out-of-pocket (OOP) payment (in KES) for public sector injectable and implant among users with non-zero expenditure by wealth quintile

| Wealth quintile | Injectable | | | Implant | | |
|---|---|---|---|---|---|---|
| | n | Mean (95% CI) | Quintile ratio* | n | Mean (95% CI) | Quintile ratio* |
| Poorest | 147 | KES 65 (59 to 72) | | 27 | KES 267 (146 to 389) | |
| Poor | 247 | KES 61 (57 to 65) | | 76 | KES 231 (200 to 262) | |
| Middle | 256 | KES 66 (62 to 70) | | 68 | KES 253 (205 to 301) | |
| Wealthy | 197 | KES 66 (61 to 72) | | 78 | KES 317 (187 to 447) | |
| Wealthiest | 116 | KES 78 (69 to 87) | 1.2 (p=0.033) | 82 | KES 240 (190 to 291) | 0.90 (p=0.660) |

*Ratio of mean OOP expenditure comparing the wealthiest users with the poorest users. Adjusted Wald-type test based on the delta method was used to test for significance.
1 KES=US$0.0114.
KES, Kenyan shillings.

for injectable and implant. In 2010, Kenya's Ministry of Health devolved decision-making power and budgets to the county level, although policy continued to be set at the national level.[32] Despite recent gains in national modern contraceptive prevalence and reduction of unmet need, large regional disparities in coverage remain.[19] Differences in regional levels of free FP, with a substantially higher proportion of users in Nairobi reporting free FP compared with nearly all other regions, suggest that counties may be operating different systems of payment for contraception or distribution channels for FP commodities. Public primary care facilities in Kenya have long faced challenges of resource scarcity.[33] As public primary care facilities cannot directly charge for FP under the current policy, when faced with declining revenue, they may introduce indirect charges, framed as registration fees or other costs, to recoup expenses.[11] Efforts to reimburse primary care facilities to account for the abolishment of user fees have been at relatively low levels, and as our findings also show, user fees above those set in national policy continue to be charged.[33] Further research is needed to understand subnational implementation of the national FP policy, the impact of facility-level strategies to cope with financial shortfalls on user's access to care and the reasons users are charged for contraceptive services.

The considerable variation in free services by method in public facilities possibly reflects differing auxiliary costs associated with dispensing methods, with, for example, more staff time, training and medical equipment required to insert IUDs and implants compared with condoms, which are often available without a consultation. We found that injectable users were significantly less likely to report receiving this method for free compared with long-acting IUD and implants or even the pill, raising questions about the long-term cost burden to users, who require resupply every 3 months for continued coverage, for this popular method.

## Limitations

This study was limited in relying on the accuracy of women's self-report of their method, source and cost of FP. While current injectable users needed to recall how much was paid up to 3 months earlier, some current implant users were asked to report the amount paid up to 3 years prior to interview, although median length of implant use was <17 months. Additionally, we were only able to consider cost and source among women who were current users of FP. Findings are likely not generalisable to former implant or injectable users, particularly if they discontinued due to costs associated with obtaining their method of choice, or to prospective users who were discouraged from initiating FP due to costs associated.

The DHS question regarding contraceptive cost asked for the total paid for commodity and consultation, and it did not capture costs associated with time and travel to obtain the method. These may be significant, particularly for rural users. We were unable to estimate the share of OOP payment for FP from total income because the DHS does not collect information on household/individual income or expenditures. As such, we cannot draw conclusions about the extent to which the amount paid for FP represents an undue burden on individual users. Additionally, DHS household wealth quintiles may not align with the poverty definition used to determine FP fee waivers or offer sufficient nuance to distinguish very disadvantaged households.[34]

We acknowledge that the first consultation visit to initiate the contraceptive method may be longer, involving counselling and taking of medical history, than a resupply visit and could result in increased cost. However, we compared results for initiators, users starting injectable and implant <3 months and 3 years, respectively, before the survey where the cost paid likely refers to the initiating consultation, against resupply users, those starting the method >3 months or 3 years prior to interview. Yet we found initiating users reported slightly lower mean costs

than resuppliers, although differences were not significant (results not shown).

Finally, FP budget implementation is done at the county-level in Kenya, yet the DHS FP cost question was intended to be representative at national, urban/rural and regional levels only[19] and thus county-level results could not be examined.

## CONCLUSIONS

Removing or subsidising costs for the poor is a core component of an equitable system of user fees for healthcare, yet our findings highlight that the poorest contraceptive users in the public sector were as likely to pay for FP services as wealthier users. Kenya's National Reproductive Health Strategy (2009–2015) outlined pro-poor strategies and objectives to increase equity of FP access. The Kenyan government has made important progress in expanding FP access but more attention is needed for implementation of user fee policies, particularly to ensure the poorest receive affordable services and to account for geographic variation, ensuring recent efforts to reimburse facilities for lost user fee revenue are done at appropriate levels. However, public sector resources alone are unlikely to meet Kenya's growing demand for modern contraception. Policymakers should consider how government resources could be targeted at those least able to tap the private sector for FP care. While individual price discrimination offers one route to targeting public services to the poor, efforts could also focus on resources—including outreach campaigns about patients' rights and correct fees—towards facilities in poor areas or towards increasing choice of affordable methods and accessible, high-quality providers for the poor. Fulfilling the promise of equity in FP access in Kenya demands turning policy intention into sustainable action from the national to facility level.

**Contributors** ER designed the research question, analysed data and prepared the manuscript. LB, MLD, CL and JB contributed to the design of the study. KLMW contributed to analysis of the data. LB, MD, FLC and EB assessed interpretation of findings and contributed to manuscript revisions. JB, TA, KLMW and ML-A reviewed and edited the manuscript. All authors read and approved the final manuscript.

**Funding** The research in this publication was supported by funding from MSD, through its MSD for Mothers programme. MSD for Mothers is an initiative of Merck & Co, Kenilworth, New Jersey, USA. EB is supported by a Wellcome Trust research training grant (#107527).

**Disclaimer** MSD had no role in the design, collection, analysis and interpretation of data, in writing of the manuscript or in the decision to submit the manuscript for publication. The content of this publication is solely the responsibility of the authors and does not represent the official views of MSD.

**Competing interests** None declared.

**Patient consent for publication** Not required.

**Ethics approval** The DHS receive government permission and follow ethical practices including informed consent and assurance of confidentiality. The Research Ethics Committee of the London School of Hygiene and Tropical Medicine approved our secondary data analysis.

**Provenance and peer review** Not commissioned; externally peer reviewed.

**Data sharing statement** The data that support the findings of this study are owned by the Demographic and Health Surveys (DHS) Program, operated by ICF International. Restrictions apply to the availability of these data, which were used under licence for the current study, and so are not publicly available. Data are available for free from the DHS Program website and available for researchers who apply for and meet the criteria for access. Legal access agreements do not allow the sharing of datasets to unregistered researchers.

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
