## [Reviewer comments · BMJ Open]

ARTICLE DETAILS

TITLE (PROVISIONAL)	Who pays and how much? A cross-sectional study of out-of-pocket payment for modern contraception in Kenya
AUTHORS	Radovich, Emma; Dennis, Mardieh; Barasa, Edwine; Cavallaro, Francesca; Wong, Kerry; Borghi, Josephine; Lynch, Caroline; Lyons-Amos, Mark; Abuya, Timothy; Benova, Lenka

VERSION 1 – REVIEW

REVIEWER	Dr Abiodun Adeniran University of Ilorin, NIGERIA
REVIEW RETURNED	14-Mar-2018

GENERAL COMMENTS	The study attempted to address an important question in reproductive health and the effort is commendable  1. Background: this is too lengthy. Authors should reduce the volume to about 50% maximum 75%- state the research question and why this is appropriate. in this case previous attempts at making contraception available and free will suffice together with the challenges. 2. Methodology: Describe in detail how you progressed from the study design to the regions, provinces, choice of individual health facilities and each participant. the sampling method need a clearer description. 3. Discussion: It was okay to give a brief summary of important findings in the first paragraph as done by authors. However, there was scanty discussion in the body of the discussion. It should be made more robust by brining to the fore important aspects. the references used for comparison were too few, the discussion was not robust as expected. 4. Conclusion: Your conclusion must be derivable from your study- references are not expected here- it is a direct product of your work. 5. References: Majority of the references were not complete- e.. books without names of publishers etc. check references 2,8,11,14,17,18,19,20,21,22,23,25,26,27,29,30,31,33.
--

REVIEWER	Soumya Alva John Snow Inc. (JSI) United States
REVIEW RETURNED	05-May-2018

GENERAL COMMENTS	Overall this is a well written paper on a relevant topic in the current context of discussions relating to Universal health coverage. The paper lays out the changes in the reproductive health strategy in Kenya, the latest changes resulting in provision of free care in public health facilities. There is some reference to FP clients being charged for services despite the 10/20 policy established in 2004. However,
--

	given the recent changes in the policy resulting in free services, it is not clear why and how users are being charged for services in public facilities. Clarifying this in the text would be helpful as there are several analysis sections that refer to OOP payments overall and within the public sector. Are payments for those who were charged for services under the pre-2013 policy or are these individuals who made payments post 2013 as well? Would be good to clarify reasons for payment at public facilities. Is it because the new policy had not fully come into effect (as mentioned in lines 23-27 on Page 5) or are there other reasons? There is some reference to this in the discussion section but would be useful to mention earlier in the paper. The section on data source under Methods (Line 26-31) refer to the use of the 2014 Kenya DHS data. In Page 4, Line 27, it is stated that only half the women were administered a women's questionnaire. This section further states that the women's questionnaire did not ask questions on the amount paid for contraceptives and so were excluded. It is not clear who has been excluded and who has not, and who was asked questions on cost of contraceptive use and who was not. Pls. clarify. Page 4 Line 34 refers to reference 24 – its not clear why this reference is listed at the current location. Is it to list what methods are included in modern methods? Would be useful to list the methods included under modern methods. Page 5 Line 12 refers to adjusted Wald tests to compare proportions – but its not clear on what proportions are being compared. Page 6 Line 30 also refers to those who made payments. As stated earlier would be good to know reasons for such payment when the policies had changed. This percentage of those who paid post 2013 seems quite high – as shown in Table 1 and Supplementary Table 3. Page 8: Table 2. Is the analysis restricted only to those who seek modern contraceptive services from government primary providers or from all providers? If only from government primary providers, it would be useful to specify this in the table where Modern method users (n=2079) is listed. Would also be useful to calculate predicted probabilities using the adjusted model for high and low wealth quintile groups and maybe also in relation to methods used (implants, injectables and IUDs). Note: The supplementary tables and figures were very hard to read.
--	---

VERSION 1 – AUTHOR RESPONSE

Reviewer 1: Dr. Abiodun Adeniran, University of Ilorin

Reviewer comments	Response
The study attempted to address an important question in reproductive health and the effort is commendable	Thank you very much for taking the time to review this paper and providing helpful comments to improve its framing and interpretation.

Background: this is too lengthy. Authors should reduce the volume to about 50% maximum 75%- state the research question and why this is appropriate. in this case previous attempts at making contraception available and free will suffice together with the challenges.	Thank you for this feedback. We have reduced the length of the Background section in response to your comments, cutting 200 words. However, we feel it is important to put Kenya's recent policy efforts and the focus of this paper in context with global family planning debates on the role of cost in contraceptive use and on targeting within total market approaches. We hope the results of this paper contribute to these debates both within Kenya and more broadly and shed light on the understudied subject of user fees for contraception.
Methodology: Describe in detail how you progressed from the study design to the regions, provinces, choice of individual health facilities and each participant. the sampling method need a clearer description.	Thank you for this suggestion. We have clarified under "Data Source" that the sampling strategy for the secondary data used in this analysis can be found in the Kenya 2014 Demographic and Health Survey report.
Discussion: It was okay to give a brief summary of important findings in the first paragraph as done by authors. However, there was scanty discussion in the body of the discussion. It should be made more robust by bringing to the fore important aspects. The references used for comparison were too few, the discussion was not robust as expected.	Thank you for this feedback. We have modified the Discussion section based on your comment to include more comparisons from our findings and a more robust discussion of how our findings fit in to the larger family planning policy context and challenges in Kenya and broader conceptual approaches to expanding contraceptive access. In particular, we have highlighted what we think is a key study finding around the lack of difference in the proportion of users reporting free family planning from public providers between the poorest and wealthiest quintiles and what this means for current government approaches to expanding equitable access to affordable family planning care. We hope the changes made address your concerns.
Conclusion: Your conclusion must be derivable from your study- references are not expected here- it is a direct product of your work.	Thank you for this comment. We have modified the conclusion to remove references and to clarify the points derived from our findings.
References: Majority of the references were not complete- eg. books without names of publishers etc. check references 2,8,11,14,17,18,19,20,21,22,23,25,26,27,29,30,31,33.	Thank you for this suggestion. The references mentioned are for reports, working papers and other grey literature sources and websites. We have updated these references to ensure a publisher is listed where required. We are using BMJ Open referencing formats, which does not include the web address for reports or working
	papers in citation lists, but we are happy to additionally include web links to available PDF versions of these documents to ensure readers are able to locate the relevant references.

Reviewer 2: Soumya Alva, John Snow Inc. (JSI)

Reviewer comments	Response
Overall this is a well written paper on a relevant topic in the current context of discussions relating to Universal health coverage.	Thank you very much for taking the time to review this paper and providing helpful comments.
The paper lays out the changes in the reproductive health strategy in Kenya, the latest changes resulting in provision of free care in public health facilities. There is some reference to FP clients being charged for services despite the 10/20 policy established in 2004. However, given the recent changes in the policy resulting in free services, it is not clear why and how users are being charged for services in public facilities. Clarifying this in the text would be helpful as there are several analysis sections that refer to OOP payments overall and within the public sector. Are payments for those who were charged for services under the pre-2013 policy or are these individuals who made payments post 2013 as well?	Thank you very much for your question and careful reading. While our paper does not aim to specifically evaluate implementation of the June 2013 policy, we use the June 2013 policy as a point of reference to understand who reports obtaining their FP method for free and from which type of public sector provider. This analysis is based on the Kenya 2014 DHS survey which collected data between May-October 2014. Reported user charges for FP in public health centres and dispensaries therefore mean that user fees are still being charged contrary to the 2013 user fee removal policy. However, users of long-term methods (IUD and implant) could have obtained their method before June 2013, and we disaggregate those users by whether they initiated use before/after the June 2013 policy in Supplementary Table 3. Otherwise, users obtained their short-term method after June 2013 (it is possible a user could have stockpiled pills or condoms from before June 2013 until their date of interview in May-October 2014, but we find this highly unlikely). We agree that it is not clear why a substantial proportion of users are being charged for family planning services from public facilities, especially public primary care providers. While we suggest some potential reasons in the Discussion, including facilities using indirect charges, such as ‘registration fees’ to try to recoup expenses, we believe this is an area for further policy implementation research that is beyond the scope of this analysis.
Would be good to clarify reasons for payment at public facilities. Is it because the new policy had not fully come into effect (as mentioned in lines 23-27 on Page 5) or are there other reasons? There is some reference to this in the	Thank you for this suggestion. Unfortunately, individuals reporting payments from public sector providers, including at all three levels of public facilities, are not asked for the reason for payment. The DHS question, asked of relevant

discussion section but would be useful to mention earlier in the paper.	method users regardless as to the source reported, reads: “The last time you obtained (METHOD), how much did you pay in total, including the cost of the method and any consultation you may have had.” We agree that it is not clear why users of public providers, particularly of public primary care providers, would be reporting payment in light of the free services policy. As we mention in the Discussion, this may relate to sub-national implementation of the June 2013 policy and is an area for further research.
The section on data source under Methods (Line 26-31) refer to the use of the 2014 Kenya DHS data. In Page 4, Line 27, it is stated that only half the women were administered a women’s questionnaire. This section further states that the women’s questionnaire did not ask questions on the amount paid for contraceptives and so were excluded. It is not clear who has been excluded and who has not, and who was asked questions on cost of contraceptive use and who was not. Pls. clarify.	Thank you for this suggestion and request for clarification. We have amended the section on “Data source” to clarify that women in a random half of the households were administered a short-version Woman’s Questionnaire. This shorter questionnaire did not ask about amount paid for contraception. The long-version Woman’s Questionnaire did include questions on cost of contraception, so we included only those women who were administered the long-version questionnaire.
Page 4 Line 34 refers to reference 24 – its not clear why this reference is listed at the current location. Is it to list what methods are included in modern methods? Would be useful to list the methods included under modern methods.	Thank you for this question. We have amended the text to clarify that we are using the Hubacher & Trussell definition of modern methods. We have omitted the full list of modern methods in order to meet word count limitations.
Page 5 Line 12 refers to adjusted Wald tests to compare proportions – but its not clear on what proportions are being compared.	Thank you for this feedback. We have amended the sentence to read: “Adjusted Wald tests were performed to compare proportions reporting free FP by facility level and user characteristics.”
Page 6 Line 30 also refers to those who made payments. As stated earlier would be good to know reasons for such payment when the policies had changed. This percentage of those who paid post 2013 seems quite high – as shown in Table 1 and Supplementary Table 3.	Thank you for this suggestion. We absolutely agree that it would be good to know the reasons for payment. Unfortunately, as noted above, the DHS questionnaire does not ask respondents for the reason for the charge. We have amended the Discussion to call for future research to understand more about these payments.
Page 8: Table 2. Is the analysis restricted only to those who seek modern contraceptive services from government primary providers or from all providers? If only from government primary providers, it would be useful to specify this in the table where Modern method users (n=2079) is listed. Would also be useful to calculate predicted probabilities using the	Thank you for the question and suggestion. We have amended Table 2 to clarify that the model is restricted to only modern contraceptive users whose source was a public primary care provider (ie government dispensary or government health centre).

adjusted model for high and low wealth quintile groups and maybe also in relation to methods used (implants, injectables and IUDs).	We agree that predicted probabilities can offer a more intuitive way of understanding differences compared to odds ratios. However, we feel that in this case, predicted probabilities would not significantly enhance the larger point about the lack of differences in the adjusted model in the odds of free family planning across the wealth quintiles.
Note: The supplementary tables and figures were very hard to read.	Thank you for this comment. We have reformatted the supplementary tables to improve legibility.

VERSION 2 – REVIEW

REVIEWER	DR ABIODUN ADENIRAN UNIVERSITY OF ILORIN/ NIGERIA
REVIEW RETURNED	18-Jul-2018

GENERAL COMMENTS	The previously highlighted corrections have been implemented. Baring other minor corrections that will be taken care of during editing, I am satisfied with article.
--

REVIEWER	Soumya Alva John Snow Inc, USA
REVIEW RETURNED	13-Aug-2018

GENERAL COMMENTS	Thanks for addressing the comments and issues previously raised and the revised manuscript. A comprehensive analysis is presented. Here are a few additional comments. The main comment relates to the various sample sizes for the different analyses and adding a little explanation on the numbers and choice of sample for each analysis. Version with track changes.  • Methods under Data source: Page 27 of 41 Line 35. The information provided is a little repetitive. The sentence “women in a random sample... contraceptive method” can be removed. Instead the next sentence can be modified to say: “We include in our analysis women in half of the randomly selected households who were administered the long version Women’s questionnaire, which included a question.... for the payment”. • Would be useful to know how many women fell in this category. • Page 28 of 41 line 19. “Contraception on the 2014 Kenya DHS” should be “Contraception in the 2014 Kenya DHS” • Page 28 of 41 Line 23 – Section “Analysis of free or “registration fee only” FP in the public sector. Would be useful to clarify why only primary care was included in this analysis. • Results section: Page 29 of 41 line 21. The different
---

	analyses presented in the different sub sections under results cover different populations and it tends to get confusing on how many women we are talking about and why a different sample has been chosen would be useful. Some clarification on the different sample sizes for the different analyses would be helpful. For example. Table 2 is restricted to primary care facilities only and a clarification statement to that effect would be useful.  • Would help to know sample sizes for public/private facility, and those with cost data perhaps even broken down by method. Some info on the method mix would also be useful –the numbers for injectable and implants are provided. However the analysis covers other modern methods as well. • Maybe the results section could start with a table showing these numbers, which would make reviewing the subsequent analyses easier. • Table 2 – pls. explain why the odds ratios for condoms are so high • Table 4 – not clear why the totals for each sub category – wealth quintile. Residence, educational attainment, age group, region. One would assume that these variables had no missing data. • In Table 4, may be useful to show numbers separately for the public and private sectors? Is there a reason this was not done? Page 36 of 41, lines 26-27 should say “As public primary care facilities cannot directly charge for FP under the current policy” Line 32 should say “implementation of the national FP policy”
--	---

VERSION 2 – AUTHOR RESPONSE

Reviewer comments	Response
Methods under Data source: Page 27 of 41 Line 35. The information provided is a little repetitive. The sentence “women in a random sample... contraceptive method” can be removed. Instead the next sentence can be modified to say: “We include in our analysis women in half of the randomly selected households who were administered the long version Women’s questionnaire, which included a question.... for the payment”.	Thank you for this helpful, more concise suggestion. We have made the change as requested.
Would be useful to know how many women fell in this category.	Thank you for this feedback, we have added the sample size (n=14,741) to the sentence as modified above.
Page 28 of 41 line 19. “Contraception on the 2014 Kenya DHS” should be “Contraception in	Thank you, we have corrected the typo.

the 2014 Kenya DHS”	
Page 28 of 41 Line 23 – Section “Analysis of free or “registration fee only” FP in the public sector. Would be useful to clarify why only primary care was included in this analysis.	Thank you for this request for clarification. On Page 28, lines 26-27 on the earlier revised version with tracked changes, we have indicated in brackets that public primary care providers were included in this analysis as they were subject to the June 2013 fee abolishment policy. We included public hospitals in the descriptive results (Table 1), simply as a point of comparison.
Results section: Page 29 of 41 line 21. The different analyses presented in the different sub sections under results cover different populations and it tends to get confusing on how many women we are talking about and why a different sample has been chosen would be useful. Some clarification on the different sample sizes for the different analyses would be helpful. For example. Table 2 is restricted to primary care facilities only and a clarification statement to that effect would be useful.	Thank you for this feedback. We have added a clause to the description of Table 2 to clarify that this analysis presents results for users of public primary care facilities, which were subject to the June 2013 fee abolishment policy, and Table 2 shows the sample size included in the analysis (n=2079).
Would help to know sample sizes for public/private facility, and those with cost data perhaps even broken down by method. Some info on the method mix would also be useful – the numbers for injectable and implants are provided. However the analysis covers other modern methods as well.	Thank you for this suggestion. The overall method mix is provided in Figure 1a, and the sample sizes for public and private provider types can be seen via the total column in Figure 1b. Based on this feedback, we have also modified Figure 1b to show the sample sizes for each wealth quintile and for the total column to make this clearer – these are the same values as in Figure 1a (there are no missing data for respondent’s wealth quintile). However, as not all modern method users were asked to report the cost to obtain the method, we have added an additional supplementary
	table with the sample sizes by provider type and method mix by provider (Supplementary Table 3) to further clarify the sample sizes within sub-categories.

Maybe the results section could start with a table showing these numbers, which would make reviewing the subsequent analyses easier.	Thank you for this suggestion. We acknowledge that the paper presents results of several different analyses of sub-populations of respondents, owing to the question skip patterns used by the DHS and in the relevance to Kenya's recent family planning policy. We have modified the paper in response to your feedback to further clarify the sample sizes of different methods by provider type by adding a supplementary table (Supplementary Table 3) and in clarifying within the manuscript text as to why a sub-population has been included in a particular table or analysis. We feel the changes made address the concern with the flow of analyses and sub-populations under consideration.
Table 2 – pls. explain why the odds ratios for condoms are so high	The large OR for condom users in Table 2 is a reflection of the very large percentage of condom users (>90%) reporting obtaining this method for free from all public sector provider categories, shown in Table 1. In the Discussion section (fifth paragraph) we highlight that the considerable variation in free FP by method may reflect different auxiliary costs associated with dispensing the method and that condoms are often available without consultation, thus more likely to incur no fee.
Table 4 – not clear why the totals for each sub category – wealth quintile. Residence, educational attainment, age group, region. One would assume that these variables had no missing data.	Thank you for this question. We present the sample size for each sub-category as the distribution of injectable/implant users is not even across the socio-demographic characteristic. We want to be clear that the cost estimates presented for each sub-category are, in some cases, based on relatively small sample sizes. The reviewer is correct; there are no missing data for these socio-demographic variables.
In Table 4, may be useful to show numbers separately for the public and private sectors? Is there a reason this was not done?	Thank you for this suggestion. We chose not to further disaggregate results presented in Table 4 by source of the injectable/implant due to the small sample sizes in several provider categories. As shown in Figure 1c, use of private providers in the poorest wealth quintile, particularly among implant users, is quite low. We disaggregate cost of injectable and implant across public and private provider types in Table 3, but due to small sample sizes among

	some private provider types and sociodemographic categories, we did not think it appropriate to further sub-divide categories.
Page 36 of 41, lines 26-27 should say “As public primary care facilities cannot directly charge for FP under the current policy”	We have amended the sentence as suggested.
Line 32 should say “implementation of the national FP policy”	The grammatical error has been corrected.

VERSION 3 – REVIEW

REVIEWER	Soumya Alva John Snow, Inc.
REVIEW RETURNED	12-Nov-2018
GENERAL COMMENTS	Thanks for addressing the comments and questions.